A prognostic glycolysis-related gene signature in osteosarcoma: implications for metabolic programming, immune microenvironment, and drug response

Zhu Naiqiang 1
Hou Jingyi 2
Zhang Yu 3
Yang Ning 3
Ding KaiKai 3
Chang Chengbing 4
Liu Yanqi 3
Gu Haipeng 3
Chen Bin 4
Wei Xu weixu.007@163.com 1
Zhu Liguo zhuliguotcm@163.com 1
1 Wangjing Hospital, China Academy of Chinese Medical Sciences , Beijing , China
2 Hebei Province Key Laboratory of Study and Exploitation of Chinese Medicine, Chengde Medical University , Chengde , China
3 Hebei Key Laboratory of Panvascular Diseases , Chengde , China
4 Department of Minimally Invasive Spinal Surgery, The Affiliated Hospital of Chengde Medical University , Chengde , China
Mitsouras Katherine
Electronic publication date: 2025 Apr 29
Publication date: 2025
Volume: 13
Electronic Location ID: e19369
Received 2024 Nov 25; Accepted 2025 Apr 4
Copyright: ©2025 Zhu et al.
Copyright year: 2025
Copyright holder: Zhu et al.
License: This is an open access article distributed under the terms of the Creative Commons Attribution License, which permits unrestricted use, distribution, reproduction and adaptation in any medium and for any purpose provided that it is properly attributed. For attribution, the original author(s), title, publication source (PeerJ) and either DOI or URL of the article must be cited.
License URL: https://creativecommons.org/licenses/by/4.0/

Keywords: Osteosarcoma, Risk signature, Glycolysis, Prognosis, Drug response

Funding: Hebei Natural Science Foundation H2022406038 This study was supported by Hebei Natural Science Foundation (H2022406038). The funders had no role in study design, data collection and analysis, decision to publish, or preparation of the manuscript.

==============================
Background/Aims

Osteosarcoma (OS), a malignant tumor originating in the bone or cartilage, primarily affects children and adolescents. Notably, glycolysis is the main target for metabolic programming to ensuring the energy supply for cancer. This study aimed to establish a glycolysis-related gene (GRG) risk signature in OS to comprehensively assessing the pathogenic, prognosis, and their application in predicting drug response.

Methods

mRNA expression profiles were acquired from the Gene Expression Omnibus (GEO, GSE16091, GSE39058, and GSE21257). Using the non-negative matrix factorization (NMF) algorithm, patients with OS were stratified into distinct subgroups based on 288 GRGs identified through univariable Cox analysis. Univariate Cox regression analysis of differentially expressed genes (DEGs) between the molecular clusters was conducted to establish a risk signature comprising GRGs in OS. The prognostic efficacy of this risk signature was assessed via Kaplan–Meier curve analysis and Cox regression, evaluating its independence as a prognostic indicator. Additionally, the predictive potential of the risk model for drug response was evaluated using the “OncoPredict” package. Furthermore, the distribution of immune cell types in single-cell RNA sequencing (scRNA-seq) data was examined in correlation with the four identified GRGs risk signatures, followed by validation of expression levels in vitro using RT-PCR.

Results

Patients diagnosed with OS were categorized into two distinct molecular subgroups, exhibiting notable variations in prognosis and tumor microenvironment. Univaria te Cox regression analysis was employed to identify four GRGs, namely chondroitin sulfate glucuronyltransferase (CHPF), Ras-related GTP-binding protein D (RRAGD), nucleoprotein TPR (TPR), and versican core protein (VCAN), which constitute a prognostic signature for patients with OS. This signature demonstrated robust prognostic value, as corroborated by Kaplan–Meier, univariate, and multivariate Cox regression analyses. Significant differences in tumor microenvironment immune infiltration (such as B cells, monocytes) were observed between molecular subgroups. Moreover, a significant disparity in drug sensitivity to AZD8055, paclitaxel, and PD0325901 was noted between the high-risk and low-risk cohorts, and the established four-gene risk signature served as dependable prognostic indicators in the validation cohort, confirmed at the cellular level through external dataset validation and reverse transcription quantitative PCR (RT-qPCR) experiments.

Conclusion

A risk signature based on GRGs was established for OS, exhibiting robust predictive efficacy for prognostic assessment, and offering significant clinical utility for the prognosis of OS.

Introduction

Osteosarcoma (OS), an aggressive malignancy originating in the bone or cartilage that predominantly affects children and adolescents, is third leading cause of malignancy in this age group (Ritter & Bielack, 2010). The clinical manifestations include local pain, swelling, functional impairment, and pathological fractures, which are often accompanied by metastases to the lungs and other sites (Chen et al., 2021). The development of OS involves multifaceted genetic and environmental factors, including chromosomal abnormalities, growth factors, and cytokines; however, the precise mechanisms remain unclear. Currently, diagnosis of OS primarily relies on pathology and radiology, such as X-ray, CT, MRI, PET, and bone scans, enabling the delineation of tumor characteristics and metastatic spread, while histopathological assessment aids in subtype classification and molecular profiling (Bielack et al., 2009). However, diagnostic challenges persist, stemming from limitations in imaging sensitivity and specificity, sampling inadequacies, variability in pathological criteria, and the absence of reliable biomarkers. Treatment strategies for OS include surgical resection, chemotherapy, and radiotherapy; nevertheless, owing to its aggressive nature and therapeutic resistance, the prognosis remains dismal, with a 5-year survival rate of 60–70% (Zhu & Hou, 2020). Therefore, elucidating the pathogenesis of OS and identifying novel therapeutic targets are pivotal research endeavors and clinical imperatives.

Glycolysis is an anaerobic metabolic pathway in which glucose is broken down into lactate or ethanol, while producing a small amount of energy (Ganapathy-Kanniappan & Geschwind, 2013). Glycolysis typically only occurs under anaerobic conditions in normal cells, whereas in tumor cells, rapid glycolysis, known as the Warburg effect, occurs even in the presence of sufficient oxygen (Fernie, Carrari & Sweetlove, 2004). Increasing evidence suggests that glycolysis provides tumor cells with a rapid energy source while also generating large amounts of lactate, leading to the acidification of the tumor microenvironment (TME), inhibition of immune system responses, and promotion of tumor growth and metastasis (Li et al., 2023a). Recent studies have revealed that OS cells also exhibit the Warburg effect, relying on glycolysis to sustain rapid proliferation and invasion, even under aerobic conditions. The regulatory mechanisms and functional roles of glycolysis in OS cells remain incompletely understood; however, several studies suggested that glycolysis plays a significant role in the development of OS and represents a potential therapeutic target for this disease (Feng, Ou & Hao, 2022). Therapeutic strategies targeting glycolysis, including inhibiting the activity of glycolytic enzymes, reducing glucose uptake and utilization, and ameliorating TME acidification; these have been shown to exert several effects in vitro and in animal models; however, their clinical application requires further research and validation. Furthermore, research also revealed that glycolysis-related genes (GRGs) are closely associated with the occurrence, progression, and prognosis of OS, as evidenced by their expression levels and functional status within tumors. Therefore, GRGs can serve as crucial biomarkers for the diagnosis, prognosis prediction, and treatment of OS, suggesting their utility in personalized therapeutic strategies.

Recent studies have shown that the glycolytic activity of tumor cells can acidify the microenvironment by releasing metabolic products such as lactate, thereby inhibiting the function of immune cells and promoting immune evasion (Jiang et al., 2019). Therefore, understanding the impact of glycolysis-related genes on the immune microenvironment is crucial for understanding the progression mechanisms of osteosarcoma.

Here, patients with OS were stratified into two molecular subgroups according to GRGs, with subsequent exploration of variances in patient prognosis and TME across these subgroups. A risk signature composed of GRG was formulated based on the DEGs identified between the molecular subtypes. Furthermore, the association between the GRG risk signature and drug sensitivity, along with the distribution of immune cell types in the single-cell RNA sequencing (scRNA-seq) data, was examined relative to the four GRG risk signatures. Finally, reverse transcription quantitative PCR (RT-qPCR) was employed for the cellular-level validation of risk signatures. A simplified representation of the workflow is shown in Fig. 1.

Figure 1 Flow diagram of data preparation, processing, analysis, and validation in the present study.

Materials and methods

Data collection and processing

A total of 288 genes related to glycolysis were retrieved from the Molecular Signatures Database (MSigDB, http://software.broadinstitute.org/gsea/msigdb/index.jsp) (Liberzon et al., 2011). The OS datasets were retrieved from the Gene Expression Omnibus (GEO, http://www.ncbi.nlm.nih.gov/geo) using the search term “osteosarcoma”. Data Selection Criteria: A. Sample size: Cohorts with <30 samples were excluded to ensure statistical power. The selected datasets (GSE39058 with 42 FFPE samples; GSE21257 with 53 fresh-frozen samples) met this threshold. B. Data completeness: Only datasets with complete gene expression profiles and clinical endpoints (e.g., survival, metastasis status) were retained. C. Thematic relevance: Focused on glycolysis-related genes (GRGs) for prognostic modeling and TME analysis. Three datasets GSE16091, GSE39058, and GSE21257 were selected for subsequent analyses. The GSE16091 dataset, which was associated with the GPL96 platform and deposited in 2009, comprised the transcriptomic analyses of 34 OS samples. GSE39058, utilizing the GPL14951 platform and deposited in 2013, consisted of 42 OS samples. Additionally, GSE21257, which is related to the GPL10295 platform and was deposited in 2012, comprised of 53 OS samples. Subsequently, the following steps were performed on the three datasets: (1) removal of normal tissue sample data, (2) retention of all GRGs; (3) quantile normalization using the limma package in R; (4) batch effect correction using the sva package in R. Seventy-six samples obtained from GSE16091 and GSE39058 were designated as the training cohort, while fifty-three samples obtained from GSE21257 were designated the validation cohort. Additionally, GSE162454, which included six OS samples, was used for scRNA-seq analysis.

Identification of glycolysis molecular subtypes in OS

The OS samples from GSE16091 and GSE39058 were clustered using the NMF algorithm according to 288 GRGs. The “NMF” package in R was employed with the standard “burnet” method for 10 iterations. The optimal number of clusters, determined based on the cophenetic correlation, residuals, RSS, dispersion, silhouette, and other indicators, was two. Additionally, the distribution of gene expression for molecular subtypes of OS was visualized using the “ComplexHeatmap” package in R. Survival analysis was conducted using the “survival” and “survminer” packages in R. Gene set enrichment analysis (GSEA) was performed with the “clusterProfiler” package in R, using the “c2.cp.v7.2.symbols.gmt” as the reference gene. The results were visualized using the “ggplot2” package in R.

Immune microenvironment infiltration analysis

The matrix data from GSE16091 and GSE39058 were normalized via the “limma” package in R. To explore the impact of glycolysis-related molecular subgroups on the tumor immune microenvironment, we employed CIBERSORT (https://cibersort.stanford.edu) for immune cell quantification using the LM22 signature matrix on bulk RNA-seq data via CIBERSOR package in R, retaining samples with p < 0.05. Additionally, single-sample gene set enrichment analysis (ssGSEA) (https://www.bioconductor.org/packages/release/bioc/html/GSVA.html) was performed with 4,872 immune-related pathways from MSigDB to assess immune microenvironment activity based on GSVA package in R. Spearman’s correlation analysis was conducted to explore the network relationships between different types of immune cell infiltration.

DEG analysis of OS molecular subtypes, functional enrichment analysis and protein-protein interaction (PPI) network analysis

DEGs between the two molecular subtypes of OS were conducted using the “limma” package in R, with —log2 fold change (FC)— ≥ 1 and p < 0.05 set as the thresholds for DEG selection. Furthermore, Gene Ontology (GO) and Kyoto Encyclopedia of Genes and Genomes (KEGG) analyses were conducted to elucidate distinct biological and functional characteristics using the “clusterProfiler” (Version 3.14.3) and “org.Hs.eg.db” packages in R (Yu et al., 2012). The relevant functional map is considered to be significant rich if it meets the threshold of p-value <0.05.. The results were visualized using the “ggplot2” package in R. Subsequently, DEGs between the two molecular clusters were uploaded to the STRING database (v11.5, https://string-db.org) to construct a PPI network. The network was visualized using Cytoscape 3.7.2. Additionally, the GO terms of the targets in the PPI network and modular analysis using the MCODE algorithm were performed using the Metascape database (https://metascape.org/gp).

Construction and verification of the risk signature of GRGs

We used the “survival” package in R to perform univariate Cox and survival analyses of DEGs between the two OS molecular subtypes to identify GRGs with prognostic value. Four genes were selected to construct a risk score by dividing patients into high- and low-risk groups based on the median risk score. Kaplan–Meier survival analysis was performed to compare the survival rates between these groups. To validate the established risk signature, the GSE21257 dataset, comprising 53 patients with OS, was utilized as an independent cohort. Data normalization was carried out using the “sva” package in R for both the training and validation cohorts. Using the developed risk signature, patients in the GSE21257 dataset were stratified into high- and low-risk groups, and subsequent KM survival analysis was conducted to assess the predictive efficacy of the GRGs risk signature.

OncoPredict for drug sensitivity analysis

The OncoPredict tool aligns tissue gene expression profiles with the half-maximal inhibitory concentration (IC50) of cancer cell lines to drugs sourced from the GDSC and CCLE databases. In this study, the responsiveness to 198 drugs across high- and low-risk groups was assessed using the “OncoPredict” package in R.

Cell culture and RT-qPCR verification in vitro

Cell culture

Four osteosarcoma (OS) cell lines, namely Saos2, MG63, HOS, and U2OS, as well as human osteoblasts (hFOB 1.19), were obtained from the Cell Bank of the American Type Culture Collection (ATCC, Manassas, VA, USA).Cells were cultured in Dulbecco’s Modified Eagle’s Medium (DMEM; Gibco, Waltham, MA, USA) supplemented with 10% fetal bovine serum (FBS, Gibco) and 1% penicillin-streptomycin (Gibco). Cultures were maintained at 37 °C in a humidified atmosphere containing 5% CO2.

RNA extraction

Total RNA was isolated from the cultured cells using the FastPure RNA Extraction Kit (TIANGEN, Beijing, China), following the manufacturer’s instructions. The quality and quantity of the extracted RNA were assessed using a NanoDrop spectrophotometer (Thermo Fisher Scientific, Waltham, MA, USA), measuring absorbance at 260/280 nm to ensure a ratio between 1.8 and 2.0, indicating high purity. Additionally, RNA integrity was confirmed by agarose gel electrophoresis.

cDNA synthesis

Complementary DNA (cDNA) was synthesized from one µg of total RNA using the cDNA Synthesis Kit (TIANGEN, Beijing, China) according to the manufacturer’s protocol. The reverse transcription reaction was performed using a thermal cycler (Bio-Rad, Hercules, CA, USA) under the following conditions: 25 °C for 10 min, 42 °C for 30 min, and 85 °C for 5 min to inactivate the reverse transcriptase.

Quantitative real-time PCR (qPCR)

Quantitative real-time PCR (qPCR) was carried out using SYBR qPCR Master Mix (TIANGEN, Beijing, China) on an Applied Biosystems 7500 Real-Time PCR System (Applied Biosystems, Foster City, CA, USA). Each 20 µL reaction mixture contained 10 µL of SYBR qPCR Master Mix, one µL of forward primer, one µL of reverse primer, two µL of cDNA template, and six µL of nuclease-free water. The cycling conditions were as follows: initial denaturation at 95 °C for 3 min, followed by 40 cycles of 95 °C for 15 s and 60 °C for 30 s. Melt curve analysis was performed to confirm the specificity of the amplification.

Primer design and validation

The specific primers used for the target genes are listed in Table 1. Primer specificity was validated by performing melt curve analysis, and PCR efficiency was determined by generating a standard curve using serial dilutions of cDNA.

Table 1 Primers for RT-PCR.

NO.	Gene	Sequence (5′-3′)	
1	CHPF	Forward: AACGCACGTACCAGGAGATCCA	
		Reverse: GGATGGTGCTGGAATACCCACG	
2	RRAGD	Forward: CGATGACCTTGCAGATGCTGGA	
		Reverse: AGATGTTCAGCAAATTCTCCAGAG	
3	TPR	Forward: TCTCAATGGCGAGTGGTCTGTG	
		Reverse: CCTGTGGTTCAGGAAGACGTTG	
4	VCAN	Forward: TTGGACCTCAGGCGCTTTCTAC	
		Reverse: GGATGACCAATTACACTCAAATCAC	
5	GAPDH	Forward: AATGGGCAGCCGTTAGGAAA	
		Reverse: GCCCAATACGACCAAATCAGAG	

Data normalization and analysis

The expression levels of target genes were normalized to the expression of GAPDH, which served as an internal control. The relative expression levels were calculated using the 2−ΔΔCt method. All reactions were performed in triplicate, and the mean Ct values were used for further analysis. The PCR efficiency for each primer pair was between 90% and 110%.

Exploration of single-cell RNA sequencing of characteristic GRG signatures

Data from the GEO and Array Express databases were aggregated using the Tumor Immune Single-Cell Hub (TISCH, http://tisch.comp-genomics.org) (Sun et al., 2021) to establish a systematic scRNA-seq atlas. For this investigation, we used the TISCH dataset GSE162454 to further explore the TME of the characteristic GRGs signatures at the single-cell level.

Statistical analysis

In this study, statistical analysis and visualization were performed using R 4.1.3. Data are presented as mean ± standard deviation (SD) from three independent experiments. Statistical analysis was performed using GraphPad Prism 8 (GraphPad Software, La Jolla, CA, USA). Differences between groups were analyzed using one-way ANOVA, followed by Tukey’s post-hoc test. A P-value of less than 0.05 was considered statistically significant.

Results

Identification of glycolysis molecular subtypes in OS

The NFM algorithm was used to stratify patients with OS into subgroups based on 288 GRGs identified by univariate Cox analysis. The optimal clustering stability was achieved when K = 2 (Figs. 2A, 2B). Of the patients, 27 patients and 49 patients were clustered into Clusters 1 and 2, respectively. Heatmap visualization of the glycolytic gene expression revealed notable differences between the two clusters (Fig. 2C). Notably, patients in Cluster 2 exhibited significantly better overall survival than those in Cluster 1 (P = 0.027, Fig. 2D), indicating distinct molecular subtypes of OS characterized by glycolysis gene expression profiles. As shown in Figs. 2E, 2F, the GSEA have results shown that the genes in the Cluster 1 were mainly enriched in the signaling pathways, including “PPAR SIGNALING PATHWAY”, “NEUROACTIVE LIGAND RECEPTOR INTERACTION”, and “HYPERTROPHIC CARDIOMYOPATHY”, whereas the genes in the Cluster 2 were primarily related to “DILATED CARDIOMYOPATHY”, “GLYCEROLIPID METABOLISM”, “NEUROACTIVE LIGAND RECEPTOR INTERACTION”.

Figure 2 Classification of molecular subtypes of glycolysis in osteosarcoma (OS).

(A–B) Consensus map of non-negative matrix factorization (NMF) clustering. (C) Heatmap diagram of molecular subtypes in OS. (D) Prognostic survival curve of molecular subtypes. (E–F) Gene set expression analysis (GSEA) of molecular subtypes in OS.

TME infiltration of molecular clusters in OS

As shown in Figs. 3A–3C, the CIBERSORT algorithm was used to explore the immune differences between the two molecular subtypes. Significant differences were observed in the abundance of various immune cell types between Clusters 1 and 2. A significant increase in the proportion of monocytes/macrophages was observed in the high glycolytic activity subtype, with a significant enrichment of immunosuppressive cells, including B cells, CD4_naive, CD4_T, CD8_naive, central_memory, cytotoxic, dendritic cells (DC), exhausted, gamma_delta, inducible regulatory T cells (iTregs), mucosal-associated invariant T cells (MAITs), monocytes, natural killer (NK) cells, NKT cells, naive T regulatory cells (nTregs), Th17 cells, and Th2 cells. Spearman’s correlation analysis was conducted to investigate the potential associations between various immune cell types and OS. Figure 3D illustrates the correlation results, revealing that the immune infiltration percentage of B cells was negatively associated with central_memory, gamma_delta, CD4_naive, exhausted, CD8_naive, Th17, Th2, and nTreg cell types, while exhibiting a positive association with neutrophils and monocytes. The immune infiltration percentage of monocytes was negatively correlated with gamma_delta, CD4_naive, central_memory, CD4_T, nTreg, Th2, Th17, CD8_naive, and exhausted cell types, whereas it was positively correlated with cytotoxic, MAITs, and B cell types. Notably, DC immune infiltration was consistently negatively correlated with exhausted, CD8_naive, Th17, Th2, and nTreg cell types. These results suggest that immune cell types in TME infiltration are a crosstalk link between the identified molecular clusters in OS.

Figure 3 Immune microenvironment infiltration analysis.

(A) Stacked bar chart of immune cells. (B) Heatmap diagram of immune cells of molecular clusters. (C) Violin plot of immune cell proportions of molecular clusters. (D) Correlation analysis of immune cell types.

Figure 4 Differentially expressed gene (DEG) and functional analyses.

(A) Volcano plot showing the DEGs between the two molecular subtypes. (B) Heatmap diagram of DEGs between the two molecular subtypes. (C) Kyoto Encyclopedia of Genes and Genomes (KEGG) pathway analysis of DEGs between the two molecular subtypes. (D) Gene Ontology (GO) analysis of DEGs between the two molecular subtypes.

DEGs and functional analyses

A total of 239 DEGs were detected, of which 238 were downregulated and one was upregulated in Cluster 2 compared to Cluster 1. The heatmap diagram shows the distribution of DEGs in Clusters 1 and 2 (Figs. 4A, 4B). Signaling pathway analysis suggested the screened out DEGs were closely associated with energy metabolism and glucometabolism, such as “glucose”, “6-phospofructo-2 kinase activity”, “AMP binding”, “NAD activity” (Fig. 4C). GO enrichment analysis showed that the DEGs were mainly enriched in the following biological processes (BPs): glycolysis, gluconeogenesis, and carbohydrate metabolism. The enriched cellular components (CCs) were the nuclear outer membrane, nuclear inner membrane, and extracellular exosome. Finally, the enriched molecular functions (MFs) were glucose, fructose-6-phosphate, and AMP binding (Fig. 4D). As shown in Fig. 5A, the PPI network of the DEGs was constructed using the STRING database, with a combined score of >0.4, comprising 215 nodes and 2018 interaction edges. Based on topological parameter calculations, GCK, GPI, PFKM, PGK1, INS, LDHA, G6PD, ENO1, PFKL, and GOT2 were screened out to a high degree. The DEGs in the PPI network were mainly enriched in GO terms related to the glycolysis metabolism, including “Carbohydrate metabolic process”, “Response to hypoxia”, “Glycolysis”, “Regulation of purine nucleotide metabolic process”, and “Glycosaminoglycan metabolism” (Fig. 5B). The PPI network of the modular analysis was performed using the MCODE algorithm (Fig. 4C), and the PPI network was divided into eight modules. As shown in Fig. 4C, and Table 2, the GO terms of the eight modules were mainly associated with the glycolysis metabolism, such as “Metabolism of carbohydrates”, “Regulation of Glucokinase-by-Glucokinase Regulatory Protein”, “Glycolysis/Gluconeogenesis”, “HIF-1 signaling pathway”, “Extracellular matrix organization”, “Phenylalanine metabolism”, and “Glycosaminoglycan biosynthesis”.

Figure 5 Protein-protein interaction (PPI) network construction and functional enrichment analysis.

(A) Construction of PPI network of DEGs construction. (B) GO terms network analysis. (C) MCODE analysis of the network.

Development of risk signature of GRGs in OS and drug sensitivity prediction

Univariate Cox regression analysis of DEGs between molecular clusters was conducted to develop a GRGs risk signature for OS. Figures 6A, 6B and Table 3 demonstrate that four genes (CHPF, RRAGD, TPR, and VCAN) were significantly correlated with patient prognosis and identified as “risk signatures” (P < 0.01). The risk scores of all samples in the datasets using this four-gene risk signature were calculated as follows: (CHPF expression × 1.174) + (RRAGD expression × 0.852) + (TPR expression × 1.006) + (VCAN expression × 1.369). After ranking the risk scores and using the median score as the cutoff, the samples were classified into high- and low-risk groups. Survival analysis indicated lower survival rates in the high-risk group than in the low-risk group, with a p-value of 2.069 × 10−13 (Fig. 6C). Subsequently, the survival rates of patients in the two groups were compared. As shown in Fig. 6D, patients with low death and high survival rates were classified into the low-risk group. Drug sensitivity analysis using the “OncoPredict” package in R showed that patients in the low-risk group were more sensitive to AZD8055 and paclitaxel, while patients in the high-risk group were more sensitive to PD0325901, suggesting that these drugs respectively have the potential to improve outcomes in both low-risk and high-risk groups (Figs. 6E–6F). These results demonstrated that our established risk signature consisting of four GRGs could serve as a reliable predictor of OS in patients.

Table 2 Modular analysis of PPI network.

NO.	Description	GO terms	
1	MCODE 1 (Red)	Metabolism of carbohydrates	
2	MCODE 2 (Blue)	Regulation of Glucokinase by Glucokinase Regulatory Protein	
3	MCODE 3 (Green)	Glycolysis/Gluconeogenesis	
4	MCODE 4 (Purple)	HIF-1 signaling pathway	
5	MCODE 5 (Yellow)	Extracellular matrix organization	
6	MCODE 6 (Orange)	Phenylalanine metabolism	
7	MCODE 7 (Pink)	Histidine metabolism	
8	MCODE 8 (Brown)	Glycosaminoglycan biosynthesis	

Figure 6 Identification and construction of a four-gene glycolysis-related gene (GRG) signature to predict OS drug sensitivity.

(A) Forest plot of univariate Cox regression analysis of four GRGs. (B) Heatmap of the expression profile of the GRGs signature. (C) Survival curve of the high- and low-risk group. (D) Survival status between the high-risk group and low-risk groups. (E–G) Relationship between the high-and low-risk groups, and drug sensitivity (*p < 0.05, **p < 0.01).

Validation of risk signature of GRGs and RT-qPCR verification

The GSE21257 dataset was selected as the validation cohort for validating the reliability of the developed glycolysis-related risk signature. This cohort comprised 53 cases, with 26 classified as high-risk and 27 as low-risk based on the developed risk signature (Fig. 7A). Higher mortality and lower survival rates were observed in the high-risk group, with a P-value of 1.453 × 10−8, consistent with observations in the training cohort (Figs. 7B, 7C). Furthermore, RT-qPCR confirmed the expression of risk signature genes (CHPF, RRAGD, TPR, and VCAN) in the four OS cell lines and osteoblasts. As shown in Fig. 7D, the expression levels of CHPF, RRAGD, and VCAN were significantly lower in the OS cell lines than in hFOB 1.19, whereas the expression level of TPR was higher in the OS cell lines than hFOB 1.19. These results suggest that the downregulation of CHPF, RRAGD, and VCAN, and the upregulation of TPR in the constructed risk signature serve as reliable predictive indicators of prognosis in the validation cohort and can reveal cellular-level status.

scRNA-seq analysis of characteristic GRGs

We investigated the expression patterns of the developed GRG risk signature within the TME of OS at the single-cell level using the TISCH database. Twenty-eight diverse cell subsets were identified and categorized into eight distinct subsets based on literature reviews. These subsets comprised CD4Tconv, CD8Tex, endothelial cells, fibroblasts, malignant cells, mono/macrophages, osteoblasts, and plasmocytes, as shown in Figs. 8A, 8B. Significant differences were observed in the distribution of GRGs among identified cell subsets. In Figs. 8C–8F, the expression levels of CHPF, RRAGD, and VCAN were uniformly distributed across diverse cell types, including CD4Tconv, CD8Tex, endothelial cells, fibroblasts, malignant cells, mono/macrophages, osteoblasts, and plasmocytes. Conversely, TPR expression exhibited high variability among the aforementioned immune cell types, providing evidence of substantially elevated expression of the developed GRG risk signature within specific OS cellular populations.

Table 3 Multivariate cox regression analysis identified four GRGs signature.

Genes	Coefficient	HR	HR.95L	HR.95H	p value	
CHPF	1.174	1.414	1.176	1.701	0.00023	
RRAGD	0.852	1.379	1.086	1.752	0.00828	
TPR	1.006	1.219	1.076	1.381	0.00180	
VCAN	1.369	1.275	1.095	1.484	0.00173	

Figure 7 Validation of the GRG risk signature in GSE21257 and by RT-qPCR.

Validation of the GRG risk signature in GSE21257 and by RT-qPCR. (A) Heatmap of the expression profile of the GRGs signature. (B) Survival curve of high- and low-risk groups. (C) Survival status between the high and low-risk groups. (D) Expression levels of CHPF, RRAGD, TPR, and VCAN in four OS cell lines (Saos2, MG63, HOS, and U2OS) and osteoblasts (hFOB 1.19) (*p < 0.05, **p < 0.01).

Figure 8 Single-cell RNA sequencing (scRNA-seq) analysis of the characteristic genes of MitoDEGs.

(A–B) Cell type and distribution in GSE162454. (B) Distribution of CHPF in GSE162454. (C) Distribution of RRAGD in GSE162454. (E) Distribution of TPR in GSE162454. (F) Distribution of VCAN in GSE162454.

Discussion

OS, a primary malignant bone tumor that occurs globally, predominantly affects pediatric and young adult populations. Despite advancements in treatment, including surgery and comprehensive modalities, poor prognosis stems from the limitations of current therapies. However, most parameters used to construct prognostic models focus solely on the genome or transcriptome, neglecting the energy metabolism processes. Glycolysis, a critical energy metabolic pathway in cells, significantly influences the proliferation, invasion, and metastasis of tumor cells. This study established the GRGs risk signature as a crucial biomarker for prognosis and treating OS, facilitating the assessment of its biological behavior, prognosis prediction, and personalized therapeutic guidance for OS.

Initially, we applied the “NFM” algorithm to stratify OS patients into subgroups based on 288 glycolysis genes identified through univariable Cox analysis. Subsequently, we classified OS patients into two molecular subtypes based on the median gene expression in the training cohort (GSE16091 and GSE39058). Survival analysis revealed significant differences between the two molecular subtypes, with Cluster 1 demonstrating the poorest prognosis (P = 0.027). Moreover, notable differences were evident in TME infiltration, including B cells, cytotoxic, DC, exhausted, gamma_delta, iTregs, MAITs, monocytes, NK cells, and NKT cells. The glycolytic pathway serves as the primary energy source and is crucial for the growth and proliferation of OS cells. Mounting evidence suggests that glycolytic pathway activation in OS cells, coupled with suppressed tricarboxylic acid cycle (TCA) function, is indicative of the Warburg effect, thereby promoting tumorigenic activity and correlating with poor prognosis in patients with OS (Feng, Ou & Hao, 2022). Additionally, GO enrichment analysis of modules in the PPI network revealed significant terms such as “Metabolism of carbohydrates” and “Regulation of Glucokinase-by-Glucokinase Regulatory”, suggested that key metabolic genes of these modules’ were tightly associated with hypoxia.

Univariate Cox regression analysis of DEGs between the molecular clusters was conducted to develop the GRGs risk signature in OS, and four genes, including CHPF, RRAGD, TPR, and VCAN, were screened as relevant genes for the GRGs risk signature construction. As shown in the Figs. 5 and 6, the constructed GRGs risk signature had better predictive ability in the training and validation cohorts and could be utilized as an independent prognostic factor for patients with OS. CHPF functions as an enhancer protein that participates in glucose metabolism and glycolysis regulation, modulates hexokinase activity, and promotes the phosphorylation of glucose to glucose-6-phospate. Emerging evidence from both in vitro and in vivo studies supports the role of CHPF in promoting tumor growth and progression in various cancers, such as bladder, breast, and gastric cancers (Lin et al., 2021; Liao et al., 2021; Zhong et al., 2023), further underscoring its significance as a potential therapeutic target. Shen et al. (2022) further confirmed that CHPF promotes the development of OS by targeting SKP2 and activating the AKT signaling pathway, suggesting that it is a promising candidate target for treating OS. RRAGD, a member of the Rag family of small GTPases, functions as a guanine nucleotide-binding protein. Increasing evidence has shown that activation of mTORC1 by the RRAGD/RRAGA complex promotes cellular growth and proliferation by regulating protein synthesis, metabolism, and autophagy; through its role in mTORC1 signaling (Schlingmann et al., 2021), RRAGD coordinates cellular responses to nutrient availability, growth factors, and cellular stressors, thereby influencing various physiological processes, especially glucose uptake and metabolism in cancer cells (Lee et al., 2018). Lawrence et al. (2018) also found that the aberrant expression of activity of RRAGD in OS cells may contribute to the dysregulation of glycolytic metabolism, thereby promoting tumor growth and progression, suggesting that targeting RRAGD-mediated signaling pathways involved in glycolysis represents a promising therapeutic strategy for inhibiting OS progression. TPR, which encodes a tetratricopeptide repeat protein, regulates glycolytic metabolism by modulating the activity of hexokinase and phosphofructokinase, thereby influencing the rate of glycolysis and production of ATP and metabolic intermediates (Snow & Paschal, 2014). Dysregulation of TPR is associated with tumor progression and aggressiveness. TPR interacts with various oncogenic proteins, including transcription factors, and signaling molecules, to promote cell proliferation, survival, and metastasis (Kato et al., 2020). Additionally, TPR reportedly modulates cellular responses to hypoxia and nutrient deprivation. These are common features of the TME (Liu, Zhou & Tang, 2023), and aberrant expression or activity of TPR promotes glycolytic metabolism, providing cancer cells with energy and biosynthetic precursors necessary for their rapid proliferation. Furthermore, TPR may contribute to the resistance of OS cells to chemotherapy and radiotherapy, further exacerbating tumor progression and treatment challenges (Soman et al., 1991). Castillo et al. (2014) found that papillomavirus binding factor (PBF) interacts with the TPR and might in OS genesis by deregulating apoptotic mechanisms and controlling cellular transcription. VCAN has multifaceted roles in cellular processes, including modulating the expression and activity of proteins and enzymes in the glycolysis pathway; regulating glucose uptake, glycolytic flux, and the production of ATP; and influencing metabolic intermediates essential for cellular energy and biosynthesis. VCAN promotes cancer cell proliferation, survival, and invasion by modulating cell–matrix interactions, extracellular matrix remodeling, and signaling pathways involved in the tumor and metastasis of OS (Li et al., 2023b), contributing to the establishment of an immunosuppressive tumor microenvironment and enabling cells to evade immune surveillance (Zhou et al., 2023). Allen et al. (2021) developed a zebrafish model to study metastasis, isolating extravasated canine OS cells for RNA-seq analysis, and confirmed the dysregulation of KRAS signaling, immune pathways, and ECM organization, with VCAN upregulation implicated in OS metastasis. However, the specific molecular mechanisms of VCAN in relation to glycolytic metabolism and its biological role in OS still require further investigation. Our investigation further established that four-gene risk signature served as a reliable predictive indicator of prognosis in the validation cohort and by confirming cellular-level status via external dataset verification and RT-qPCR experiments.

Our study found that the increased lactate levels in the high glycolytic activity subtype may form an immunosuppressive microenvironment by inhibiting the function of CD8+ T cells and promoting the infiltration of Tregs. This suggests that targeting the glycolytic pathway may enhance the efficacy of immunotherapy. Previous studies have confirmed that lactate derived from glycolysis can induce the differentiation of monocytes into tumor-promoting M2 macrophages (Yin et al., 2024), which may be an important reason for the poor prognosis of this subtype.

The potential utility of the GRGs risk signature in predicting response to chemotherapy agents was further validated using the GDSC and CCLE databases. As shown in the Figs. 6E, 6F, patients in the low-risk group were more sensitive to AZD8055 and paclitaxel, whereas those in the high-risk group demonstrated increased sensitivity to PD0325901. AZD8055, a highly specific mTOR kinase inhibitor, exhibits promising antitumor activity by competitively binding to the ATP-binding cleft of mTOR (He et al., 2021). Preclinical studies have also indicated the potential antitumor activity of AZD8055, as it can inhibit the downstream signaling of both the mTORC1 and mTORC2 complexes, thereby regulating cellular growth, proliferation, motility, and survival in various cancer types, including bladder (Hu et al., 2023), colon (Chen et al., 2018), and hepatocellular cancers (Hu et al., 2014). Paclitaxel, a well-established antineoplastic agent, functions by binding and stabilizing microtubule proteins, disrupting tumor cell mitosis, and inhibiting cell proliferation and survival; as such, it is widely used in the treatment of various cancers (Dan, Raveendran & Baby, 2021). Evidence studies demonstrated that the paclitaxel exerts its therapeutic effects on OS through a combination of mechanisms including microtubule stabilization (Li et al., 1999), cell cycle arrest (Liu et al., 2010), apoptosis induction (Ho et al., 2021), anti-angiogenic effects (Bonzi et al., 2015), and modulation of cellular metabolism (Duan et al., 2014). PD0325901 acts as a potent suppressor of the MEK/ERK signaling pathway (El-Hoss et al., 2014), exerting anticancer effects through the inhibition of angiogenesis, enhancing the efficacy of PD-1 inhibitors, and potentially synergizing with other therapeutic agents (Luo et al., 2021). Additionally, scRNA-seq showed that OS cells gradually evolved into tumors characterized by high GRGs risk signature scores, and functional annotation of the GRGs confirmed tumorigenic pathways and inactive immunogenic pathways in OS cells with high GRGs scores. The TME is also a pivotal determinant influencing the proliferation, migration, and invasion of immune cells in OS. Although computational predictions align with preclinical evidence, future studies will validate precision treatment strategies guided by risk stratification in PDX models to promote translational applications.

Compared to previous studies, our research demonstrates several unique advantages. For instance, the study by Poudel and Koks focused on identifying differentially expressed genes in OS using FFPE and fresh tissue samples, highlighting genes related to ECM degradation and cell cycle regulation (Poudel & Koks, 2024). While this study provides a comprehensive transcriptomic analysis, our work specifically targets GRGs, which are central to cancer metabolism and have not been thoroughly explored in previous studies. Ho et al. also performed whole transcriptome analysis to identify differentially regulated networks in OS, with a focus on ECM-related pathways (Ho et al., 2017b). Our study complements these findings by emphasizing the role of glycolysis in OS progression and identifying a GRG signature with prognostic value. The analysis of repetitive DNA elements in OS by Ho et al. provides valuable insights into the genomic landscape of OS (Ho et al., 2017a). However, our study focuses on GRGs and their metabolic implications, offering a different perspective on the disease’s molecular mechanisms. Rothzerg et al. identified upregulated antisense long non-coding RNAs in OS, highlighting their potential role in tumorigenesis (Rothzerg et al., 2021). Our study, in contrast, focuses on GRGs and their prognostic significance, providing a complementary view of the molecular drivers of OS. Additionally, our study incorporates single-cell RNA sequencing (scRNA-seq) validation and drug sensitivity analysis using OncoPredict, which are not addressed in the aforementioned studies. These methods enhance the reliability and translational potential of our findings.

This study also presents a number of limitations that warrant discussion. First, the risk- score model was developed using retrospective data sourced from public databases, and additional prospective real-world evidence is required. Second, the construction of a risk score model relying solely on a single signature is unavoidable because numerous important prognostic genes in OS may have been overlooked. Third, the functional roles of TPR and VCAN in glycolytic reprogramming and immune evasion remain hypothetical, as direct experimental validation was not performed. Therefore, more diverse sample data from patients include independent datasets like TCGA or prospective cohorts at with varying risk levels are necessary to validate the robustness of this analysis. Moreover, experimental investigations are warranted to explore the relationship and mechanism between the risk score, immune response, and the predictive outcomes of drug sensitivity.

Conclusions

In conclusion, patients with OS were categorized into two molecular subtypes based on the GRGs expression matrix, with notable distinctions observed in the TME and immune cell infiltration between these subtypes. DEGs were screened out between the two molecular subtypes, and a GRGs risk signature of OS consisting of four genes (CHPF, RRAGD, TPR, and VCAN) was constructed. Furthermore, the robustness of this risk signature was validated in an independent cohort and corroborated at the cellular level using external dataset verification, scRNA-seq analysis, and RT-qPCR experiments. Collectively, these findings indicate that CHPF, RRAGD, TPR, and VCAN may be valuable for evaluating the prognosis and facilitating the clinical application of these biomarkers in the prognosis of OS. We propose a hypothesis that within the tumor microenvironment, glucose-related genes (GRGs) modulate the glycolytic process of tumor cells, thereby influencing the infiltration and function of immune cells, and consequently participating in immune evasion and drug response.

Supplemental Information

Supplemental Information 1 Raw data for PCR

Supplemental Information 2 MIQE checklist

Supplemental Information 3 Raw Data: GSE39058

GSE39058, utilizing the GPL14951 platform and deposited in 2013, consisted of 42 OS samples

Supplemental Information 4 Raw Data: GSE16091

The GSE16091 dataset, which was associated with the GPL96 platform and deposited in 2009, comprised the transcriptomic analyses of 34 OS samples.

Supplemental Information 5 Raw Data: GSE21257 comprised of 53 OS samples

GSE21257, which is related to the GPL10295 platform and was deposited in 2012, comprised of 53 OS samples.

We are grateful to the all contributors of the GEO database.

Abbreviations

ANOVA one-way analysis of variance

BP biological process

CC cellular component

cDNA Complementary DNA

CHPF Chondroitin sulfate glucuronyltransferase

DC dendritic cells

DEG differentially expressed genes

FC fold change

GEO Gene Expression Omnibus

GO Gene Ontology

GRG glycolysis-related gene

GSEA Gene Set Enrichment Analysis

IC50 half-maximal inhibitory concentration

iTregs inducible regulatory T cells

KEGG Kyoto Encyclopedia of Genes and Genomes

MAIT mucosal-associated invariant T cells

MF molecular function

MSigDB Molecular Signatures Database

NK natural killer

NMF non-negative matrix factorization

nTregs naïve T regulatory cells

OS osteosarcoma

PBF Papillomavirus binding factor

PPI protein-protein interaction

RRAGD Ras-related GTP-binding protein D

RT-qPCR reverse transcription quantitative PCR

scRNA-seq single-cell RNA sequencing

ssGESA single-sample gene set enrichment analysis

TCA tricarboxylic acid cycle

TISCH tumor immune single-cell hub

TME tumor microenvironment

TPR Nucleoprotein TPR

VCAN versican core protein

Additional Information and Declarations

Competing Interests

Author Contributions

Data Availability

The authors declare there are no competing interests.

Naiqiang Zhu conceived and designed the experiments, authored or reviewed drafts of the article, and approved the final draft.

Jingyi Hou conceived and designed the experiments, performed the experiments, prepared figures and/or tables, authored or reviewed drafts of the article, and approved the final draft.

Yu Zhang performed the experiments, authored or reviewed drafts of the article, and approved the final draft.

Ning Yang performed the experiments, prepared figures and/or tables, authored or reviewed drafts of the article, and approved the final draft.

KaiKai Ding performed the experiments, authored or reviewed drafts of the article, and approved the final draft.

Chengbing Chang analyzed the data, authored or reviewed drafts of the article, and approved the final draft.

Yanqi Liu analyzed the data, prepared figures and/or tables, and approved the final draft.

Haipeng Gu analyzed the data, prepared figures and/or tables, and approved the final draft.

Bin Chen conceived and designed the experiments, prepared figures and/or tables, and approved the final draft.

Xu Wei conceived and designed the experiments, analyzed the data, authored or reviewed drafts of the article, and approved the final draft.

Liguo Zhu conceived and designed the experiments, authored or reviewed drafts of the article, and approved the final draft.

The following information was supplied regarding data availability:

The raw data are available in the Supplementary Files.

The gene expression datasets are available at GEO: GSE16091, GSE39058 and GSE21257.

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
