# Peer review of "A prognostic glycolysis-related gene signature in osteosarcoma: implications for metabolic programming, immune microenvironment, and drug response"

_PeerJ, doi:10.7717/peerj.19369_

## Round 0.1 · original submission · Major Revisions

The reviewers found your manuscript interesting, however they had a number of concerns that need to be addressed. One major concern was regarding the datasets you used in your validation. Specifically, the reviewers suggested that you expand validation of the glycolysis related gene signature by including independent datasets such as from TCGA. Furthermore, there are additional datasets using human samples that were not included your study but need to be mentioned in your manuscript and justification for excluding them needs to be provided, as well as the rationale for including the datasets you used. Regarding your methods, the reviewers commented on several aspects. Specifically, more details on the preprocessing methods you used needs to be provided, a False Discovery Rate correction needs to be implemented to adjust for multiple comparisons as well as more clarification on the |log fold change (FC)| >=1 criterion you used. More details on how your functional analysis, and the deconvolution analysis were performed need to be provided. With regard to the results of your study, stronger justification for including TPR and VCAN in your prognostic signature since these have not been identified by other studies, and a more extensive comparison of your results to those of other published studies need to be provided. Lastly, a discussion of possible mechanisms for how glycolytic reprogramming functions in immune evasion and drug response needs to be included, as well as a section on limitations of your study and future directions.

Please, submit a detailed rebuttal which shows where and how you have taken all comments and suggestions into consideration. If you do not agree with some of the reviewers’ comments or suggestions, please explain why. Your rebuttal will be critical in making a final decision on your manuscript. Please, note also that your revised version may enter a new round of review by the same or by different reviewers. Therefore, I cannot guarantee that your manuscript will eventually be accepted.

Reviewer 1 ·

Basic reporting

See my comments

Experimental design

See my comments. Many problems here.

Validity of the findings

See my comments

Additional comments

This is a very interesting manuscript, and a study on the glycolysis-related gene signature of OS is needed.
The manuscript, however, needs significant work before suitable for publishing.
1. The GEO search for osteosarcoma will bring up many studies based on humans. However, the authors decided to include GSE16091, which is based on dogs, GSE39058, which is based on human FFPE samples, and GSE21257, which is based on fresh human samples. However, the authors have ignored many other studies based on humans (PMID: 38966281,29050494, 29250102,34440306,https://doi.org/10.7243/2052-7993-1-3). Why did they do it? At least, they have to mention these studies and tell readers that these studies were excluded for some reason. The current version implies that all available studies were included, but this is not the case here.

2. The threshold for P < 0.05 is not sufficient to correct for multiple testing errors. The authors have to use FDR < 0.05.
3. In addition, |log fold change (FC)| ≥ 1 is also not easy to understand. I assume it is log2FC (what it usually is), then log2FC 1 is not different because log2(1)=0. Therefore, you have included the samples with no difference between comparisons in your analysis.
4. Immune microenvironment infiltration part comes out of the blue and is not mentioned in the title or somewhere else before. The authors should state that this was also part of the study.
5. How was the functional analysis done?
6. Details for the deconvolution analysis are needed. The entire script for the analysis is necessary.
7. The authors must discuss already published works as well and compare their results to these studies (PMID: 38966281,29050494, 29250102,34440306,https://doi.org/10.7243/2052-7993-1-3).

8. At the moment, the work is presented as a unique and original study, but it is based on already existing datasets, and their inclusion is not very well justified.

Reviewer 2 ·

Basic reporting

As overall, this manuscript did follow all mentioned criteria in this section, so in my opinion that is fine with this part.

Experimental design

Th same as last section and with regard with the experiment limitations that the respected authors have mentioned this section would be fine too. This experimental design did follow its aims and goals.

Validity of the findings

1- In section 3-12 (line 3 results section) why the numbers of patients in both clusters (cluster 1 and cluster 2) have such significant difference of each other? Does it have any specific reason? Or does it back to your experimental design?
2- In line 238 does it have any specific reason that your reported such large significant difference in the number of downregulation and upregulation patients? If yes, please kindly write it down the reason?

Additional comments

As overall, it was good research, and all data did explain and also presented well organized. I suggest to your valuable journal accept this manuscript after a minor revision (include the answer of my both questions). Again, I find it that follow your journal criteria.

·

Basic reporting

The manuscript is well-written and provides a clear, structured investigation into the role of glycolysis-related genes (GRGs) in osteosarcoma (OS). It effectively highlights the significance of glycolysis, the Warburg effect, and the tumor microenvironment (TME) in OS progression and prognosis. The study’s approach is novel, combining molecular subtype classification with prognostic modeling and drug sensitivity analysis. However, minor language edits and additional contextual details in the introduction, particularly regarding the clinical challenges of OS prognosis and treatment, could further enhance the manuscript. The figures are high quality and well-integrated, though their legends could benefit from more descriptive explanations to aid interpretation.

Experimental design

The study addresses a significant gap in OS research by linking GRGs to prognosis and drug response. The use of non-negative matrix factorization (NMF) for molecular subtype clustering is appropriate, and the analysis is supported by RNA-seq data from robust datasets (GSE16091, GSE39058, and GSE21257). Immune infiltration profiling using CIBERSORT and ssGSEA adds depth to the findings by highlighting the connection between glycolysis and immune modulation. However, the validation of the GRG signature should be expanded to include independent datasets like TCGA or prospective cohorts to confirm its robustness. Additional details on data preprocessing steps, including normalization and batch effect correction, would improve reproducibility.

Validity of the findings

The identification of CHPF and RRAGD as GRGs is well-supported, reflecting their critical roles in glycolysis and metabolic regulation. However, the inclusion of TPR and VCAN in the prognostic signature requires stronger justification, particularly regarding their specific links to glycolysis and their biological relevance in OS. The observed differential drug responses to AZD8055, paclitaxel, and PD0325901 between high- and low-risk groups are compelling but require experimental validation using OS models to strengthen the study’s translational potential. Moreover, exploring the mechanisms underlying immune cell differences between molecular subtypes, particularly glycolytic reprogramming’s role in immune evasion, would provide greater insight into the TME’s dynamics.

Additional comments

While the manuscript provides valuable insights, its reliance on retrospective data limits immediate clinical applicability. Expanding on the roles of TPR and VCAN, as well as providing hypotheses linking GRGs to immune modulation and drug responses, would deepen the biological discussion. Additionally, a more detailed exploration of the limitations and future directions would enhance the overall impact of the study.

---

## Round 0.2 · accepted · Accept

Thank you for thoroughly addressing the reviewers' comments and thus greatly improving your manuscript.

Reviewer 1 ·

Basic reporting

The authors have addressed all my comments, and the manuscript can be accepted.

Good

Experimental design

Good

Validity of the findings

Good

Additional comments

The authors have addressed all my comments, and the manuscript can be accepted.